# TDLAS second harmonic demodulation based on Hilbert transform

**Junfeng Wu** [1]*, **Hanyu Chen** [1], **Guohua Kang**[1,2]*, **Xu Li**[1]

**1** College of Astronautics, Nanjing University of Aeronautics and Astronautics, Nanjing, China, **2** Wuhan National Laboratory for Optoelectronics, Wuhan, China

* wujunfeng@nuaa.edu.cn (JW); kanggh@nuaa.edu.cn (GK)

**Data Availability Statement:** The data underlying the results presented in the study are available from the github (https://github.com/awublack/Hbt-TDLAS-secondHarmonic).

**Funding:** Funder: Wuhan National Laboratory for Optoelectronics Award Number: 2019WNLOKF011

## Abstract

A demodulation method for tunable diode laser absorption spectroscopy (TDLAS) second harmonic based on the Hilbert transform is proposed in this paper. The second harmonic of the TDLAS signal can be easily obtained without a reference signal. The TDLAS signal is firstly processed by band-pass filtering, then the envelope of the processed signal is obtained with Hilbert transform. And finally, the second harmonic is extracted from the $1f$ component of the envelope. The validity of the proposed method is confirmed by simulation and verified by experiment, and the result shows that the error is acceptable under the cases of weak absorbance, which indicates that the proposed method is applicable to practical trace gas detection.

## 1. Introduction

Characterized by high sensitivity, high resolution, and fast response, tunable diode laser absorption spectroscopy (TDLAS) is extensively employed in industrial process gas analysis [1, 2], environmental monitoring [3, 4], and biomedical science [5–7]. TDLAS technology can be roughly divided into direct absorption spectroscopy (DAS) and wavelength modulation spectroscopy (WMS). In WMS, the output wavelength of laser diode is simultaneously modulated by a slowly varying scanning signal and a high frequency modulation signal. As the absorption signal is transferred to high frequency, away from the interference of low frequency noise, WMS obtain a higher signal-to-noise ratio and stronger robustness than DAS [8, 9]. In practical applications, WMS is widely used, and the second harmonic of the transmitted light intensity signal is of critical importance in the calibration of gas concentration [10, 11].

Conventionally, the second harmonic is obtained by lock-in amplifier. However, analog lock-in amplifier requires reference signal that in phase with the harmonic signal. Accordingly, complex phase shifting circuits is required, and high-precision multiplier is complicated in structure and difficult to design. Besides, electronic noise and DC shift may affect the precision of analog systems [10]. Digital lock-in amplifier also requires reference signal that generated according to the frequency and phase of harmonics. To obtain the complete harmonic signal, the most common methods are to manually adjust the reference signal phase [12] or use two orthogonal signals [13–16]. The latter method is less cumbersome and is widely used in practical measurement, but the reference signals are still essential.

Grant Recipient: Guohua Kang, Doctor's degree
The funders had no role in study design, data
collection and analysis, decision to publish, or
preparation of the manuscript.

**Competing interests:** The authors have declared
that no competing interests exist.

In this paper, a TDLAS second harmonic signal demodulation method based on double Hilbert transform (HT) is proposed. And the second harmonic of the TDLAS signal can be easily obtained without reference signal. The proposed method needs no information about TDLAS signal except modulation frequency to set the passband of the filter.

## 2. Methodology

### 2.1 Theory of TDLAS

The theory of TDLAS will facilitate the derivation of the proposed method which will be presented in the next subsection. In WMS, the output wavelength of laser diode is simultaneously modulated by a slowly varying scanning signal and a high frequency modulation signal. The wavenumber of the light emitted by the diode laser can be expressed as

$$v(t) = v_c(t) + \Delta v \cos\omega t \tag{1}$$

Where $v_c(t)$ is the center light wavenumber and it is modulated by a sawtooth. $\Delta v$ is the modulation depth of the frequency modulation.

The relationship between the transmitted light intensity after passing through the gas and the emitted light intensity is

$$
\begin{aligned}
I_t(t) &= I_o(t)\exp[-\alpha(v)] \\
&= I_o(t)\exp[-PS(T)\phi(v)CL] \\
&= I_o(t)\sum_{k=0}^{\infty} A_k\cos k\omega t
\end{aligned}
\tag{2}
$$

Where $\alpha(v)$ is the absorbance; $I_t$ and $I_o$ are the transmitted and emitted light intensity; $P$ (in atm) is the total gas pressure; $S(T)$ (in cm$^{-2}$/atm) is the line strength at temperature of $T$ (in K); $\phi(v)$ (in cm) is the line profile; C (in ppm) and $L$ (in cm) are the concentration of gas and the length of light path, respectively; and $A_k$ is the $k$-th order Fourier component of transmittance.

The modulation of laser emission frequency is achieved by changing the injection current, which actually modulates the laser frequency and light intensity simultaneously, and this produce a gas independent background signal called residual amplitude modulation (RAM). And there is a phase delay between the frequency modulation and the intensity modulation [17–19]. The intensity of the light emitted by the diode laser can be expressed as:

$$I_o(t) = \bar{I}_o(t)[1 + i_1\cos(\omega t + \phi_1)] \tag{3}$$

Where $\bar{I}_o(t)$ is the average light intensity; $i_1$ is the linear intensity modulation depth, and it is proportional to $\Delta v$; $\phi_1$ is the phase shift between frequency modulation and the intensity modulation. The nonlinearity of laser intensity modulation is considered negligible here.

In conclusion, the transmitted light intensity contains many harmonic components and can be expressed as:

$$
\begin{aligned}
I_t(t) &= \bar{I}_o(t)[1 + i_1\cos(\omega t + \phi_1)]\sum_{k=0}^{\infty} A_k(t)\cos k\omega t \\
&= I_{DC}(t) + \sum_{k=1}^{\infty} H_k\cos(k\omega t + \varphi_k)
\end{aligned}
\tag{4}
$$

Where $I_{DC}(t) = \bar{I}_o(t)\left[A_0(t) + \frac{1}{2}i_1 A_1(t)\cos\phi_1\right]$; $\varphi_k$ is the phase of $k$-th order harmonic; $H_k$ denote the amplitude of $k$-th order harmonic component of the transmitted light intensity,

which is a function of $t$. Since its frequency is much smaller than $\omega$, and to simplify the expression, it is not written as $H_k(t)$ here.

As a result of RAM, there is a strong background signal in the signal $H_1$ [17]. Generally, the magnitude of $H_1$ is much larger than other high order harmonics, and the magnitude of $H_k$ diminishes quickly as the order goes up.

## 2.2 Hilbert transform

The Hilbert transform of the signal $g(t)$, denoted as $\hat{g}(t)$, is the convolution of $g(t)$ with the signal $1/\pi t$. That is:

$$\mathcal{H}[g(t)] = \hat{g}(t) = g(t) * \frac{1}{\pi t} = \int_{-\infty}^{+\infty} \frac{g(\tau)}{\pi(t-\tau)} d\tau \tag{5}$$

Here are properties of the Hilbert transform that will be used later in the derivation.

The orthogonality of Hilbert transform: after Hilbert transform, the amplitudes of the signal remain unchanged and the phases of the spectral components are shifted by $-\pi/2$. For example, the Hilbert transform of $\cos(t)$ is $\sin(t)$.

Bedrosian theorem [20]: Consider two signals $g(t)$ and $h(t)$ with nonoverlapping spectral in the frequency domain, and the Fourier transform of $g(t)$ and $h(t)$, denoted as G($f$) and H($f$) satisfy the following conditions: When $|f|>W$, G($f$) = 0; and $|f|<W$, H($f$) = 0. Then

$$\mathcal{H}[g(t)h(t)] = g(t)\mathcal{H}[h(t)] = g(t)\hat{h}(t) = g(t)\int_{-\infty}^{+\infty} \frac{h(\tau)}{\pi(t-\tau)} d\tau \tag{6}$$

To put it another way, if a signal $s(t)$ can be expressed as the product of a low-pass signal and a high-pass signal with nonoverlapping spectra, the Hilbert transform of $s(t)$ is the product of the low-pass signal and the Hilbert transform of the high-pass signal [21].

## 2.3 Principle of TDLAS second harmonic demodulation based on Hilbert transform

The process of second harmonic demodulation method based on Hilbert transform is shown in Fig 1. The demodulated signal is actually an approximate of the second harmonic. The third step "1$f$ component extraction" in Fig 1 is actually implemented by bandpass filtering as well.

It is given by the Eq (4) that the transmitted light intensity $I_t(t)$ is as follows:

$$I_t(t) = I_{DC}(t) + \sum_{k=1}^{\infty} H_k \cos(k\omega t + \varphi_k) \tag{7}$$

$I_{DC}(t)$ is the low-frequency component in the transmitted light intensity signal. Before the processing of signal, the low-frequency component (frequency $< \omega$) and high-order harmonic (frequency $\geq 3\omega$) in $I_t(t)$ is removed by a band-pass filter. The remaining signal can be expressed as:

$$I(t) = H_1 \cos(\omega t + \varphi_1) + H_2 \cos(2\omega t + \varphi_2) \tag{8}$$

Apply Hilbert transform to $I(t)$, and get $\hat{I}(t)$. According to the properties of Hilbert transform mentioned above, the effect of Hilbert transform is to produce a $\pi/2$ phase delay in $\cos(k\omega t+\varphi_k)$, and the expression of $\hat{I}(t)$ is as follows:

$$\hat{I}(t) = \mathcal{H}[I(t)] = H_1 \sin(\omega t + \varphi_1) + H_2 \sin(2\omega t + \varphi_2) \tag{9}$$

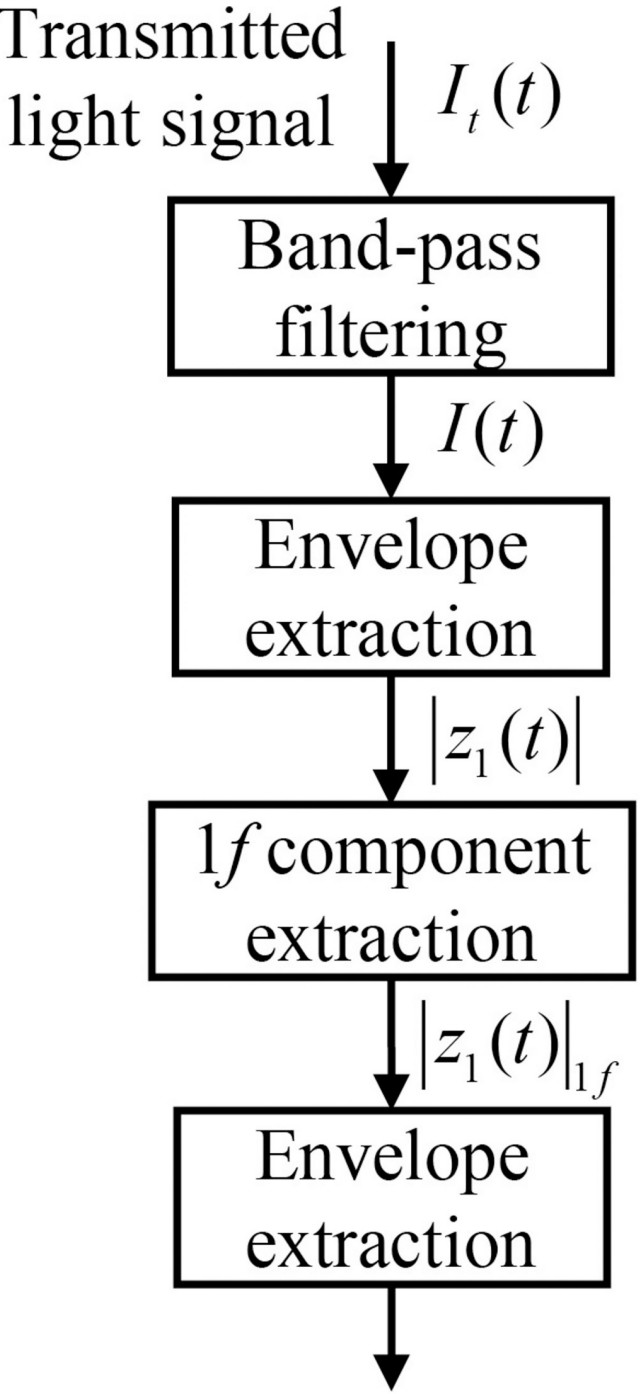

Fig 1. Flow diagram for the process of second harmonic demodulation. Envelope extraction processes are based on the Hilbert transform.

$I(t)$ and $\hat{I}(t)$ form the real and imaginary parts of an analytic signal $z_1(t)$ over the complex filed. The envelope of $I(t)$ is:

$$|z_1(t)| = \sqrt{I^2(t) + \hat{I}^2(t)} \tag{10}$$

Where $I^2(t)$ and $\hat{I}^2(t)$ is expressed as follows:

$$
\begin{cases}
I^2(t) = \sum_{k=1}^{2} H_k^2 \cos^2(k\omega t + \varphi_k) + 2H_1 H_2 \cos(\omega t + \varphi_1)\cos(2\omega t + \varphi_2) \\
\hat{I}^2(t) = \sum_{k=1}^{2} H_k^2 \sin^2(k\omega t + \varphi_k) + 2H_1 H_2 \sin(\omega t + \varphi_1)\sin(2\omega t + \varphi_2)
\end{cases}
\tag{11}
$$

Substituting $I^2(t)$ and $\hat{I}^2(t)$ into $|z_1(t)|$, we obtain:

$$|z_1(t)| = \sqrt{H_1^2 + H_2^2}\sqrt{1 + \frac{2H_1 H_2 \cos(\omega t + \Delta\varphi)}{H_1^2 + H_2^2}} \tag{12}$$

Where $\Delta\varphi = \varphi_2 - \varphi_1$.

It is to be noted that in the case of weak absorbance, $H_2 \ll H_1$. We have:

$$\frac{2H_1 H_2 \cos(\omega t + \Delta\varphi)}{H_1^2 + H_2^2} \leq \frac{2H_1 H_2}{H_1^2 + H_2^2} < \frac{2H_2}{H_1} \qquad \ll \quad 1 \tag{13}$$

Replacing $\sqrt{1 + \frac{2H_1 H_2 \cos(\omega t + \Delta\varphi)}{H_1^2 + H_2^2}}$ with its Taylor expansion, we obtain:

$$|z_1(t)| = \sqrt{H_1^2 + H_2^2}\left\{ 1 + \frac{H_1 H_2 \cos(\omega t + \Delta\varphi)}{H_1^2 + H_2^2} - \frac{H_1^2 H_2^2}{4(H_1^2 + H_2^2)^2}[1 + \cos(2\omega t + 2\Delta\varphi)] \right\} \tag{14}$$

The signal with frequency $\omega$ in $|z_1(t)|$ is easy to obtain by band-pass filtering. It is given as follows:

$$|z_1(t)|_{1f} = \frac{H_1 H_2}{\sqrt{H_1^2 + H_2^2}} \cos(\omega t + \Delta\varphi) \tag{15}$$

When $H_2 \ll H_1$, Eq (15) can be approximated as:

$$|z_1(t)|_{1f} = \frac{H_2}{\sqrt{1 + (H_2/H_1)^2}} \cos(\omega t + \Delta\varphi) \approx H_2 \cos(\omega t + \Delta\varphi) \tag{16}$$

Hence, Eq (14) indicates that for the second harmonic, the effect of the envelope extraction based on Hilbert transform is equivalent to a frequency reduction. It reduces the frequency of the second harmonic signal from $2\omega$ to $\omega$ and maintains its intensity.

As shown above, the amplitude of $|z_1(t)|_{1f}$ is approximately equal to $H_2$. The envelope of $|z_1(t)|_{1f}$, denoted by $|z_2(t)|$, can be expressed as:

$$|z_2(t)| = \sqrt{|z_1(t)|_{1f}^2 + \mathcal{H}[|z_1(t)|_{1f}]^2} = |H_2| \qquad (17)$$

The above approximation holds under the assumption that $H_2 << H_1$. Due to the strong RAM background signal in the first harmonic, in the weak absorbance condition, it is reasonable to assume that $H_2 << H_1$ [22]. The approximation in Eq (16) is satisfied well in weak absorbance conditions ($\alpha(v_0) << 10\%$). This will be verified in later simulations.

## 3. Numerical simulations

### 3.1 Simulation setup

The line shape function is described by the Lorentzian profile in the simulations:

$$\phi(v) = \frac{1}{\pi} \frac{\gamma}{(v - v_0)^2 + \gamma^2} \qquad (18)$$

Where $v_0$ (in cm$^{-1}$) is the central wavenumber and $\gamma$ (in cm$^{-1}$) is the half width at half maximum (HWHM) of the gas absorption line.

The simulation is based on the $CO_2$ absorption line at 2003.3 nm (4992.516 cm$^{-1}$), because the intensity of the absorption line of $CO_2$ is high here, we use this line to detect carbon dioxide in our experiments. The modulation index $m = \Delta v / \gamma$ is set to 2.2 to maximize the amplitude of second harmonic [23]. The parameters of the absorption line at 2003.3 nm are set according to the HITRAN database under the atmospheric pressure of 1 atm and temperature of 296 K. The line strength is 0.0306 cm$^{-2}$/atm, the $\gamma$ is set as 0.0692 cm$^{-1}$. The length of the optical path in the simulation is set as 11mm according to the optical path length of the device used in the actual experiment.

### 3.2 Signal processing and result

Global average atmospheric carbon dioxide is about 400 ppm. Therefore, the carbon dioxide concentration in this simulation is set to 400 ppm ($\alpha(v_0) = 0.0062\%$). The second harmonic is demodulated by the proposed Hilbert transform based method. Fig 2 shows the signals in the signal processing of the proposed method.

The transmitted light signal in Fig 2(A) is firstly processed by band-pass filtering, and the result is presented in Fig 2(B); secondly, the envelop of processed signal, denoted as $|z_1(t)|$, is extracted in Fig 2(C); then, the $1f$ component of $|z_1(t)|$ is obtained by band-pass filtering the $|z_1(t)|$ and presented in Fig 2(D).

The transmitted light signal in Fig 2(A) consists of low-frequency sawtooth wave and high-frequency harmonics (the figure shows the signal for only half of the scanning period), which are the common results of the intensity modulation of injection current and nonlinear absorption; In Fig 2(B), all signals except the first harmonic ($1f$) and second harmonic ($2f$) are filtered out, corresponding to Eq (8); Fig 2(C) is the envelope of the signal in Fig 2(B). Since the intensity modulation leads to a strong gas-independent RAM signal in the first harmonic, the envelope change in the figure is not obvious, the gas related signal can be seen by zooming on the vertical axis and looks like a lying "S" plus some high-frequency component. These high-frequency component is the second harmonic signal that is transformed to $1f$, corresponding to Eq (16); Then the $1f$ component of $|z_1(t)|$ is obtained by band-pass filtering and presented in Fig 2(D).

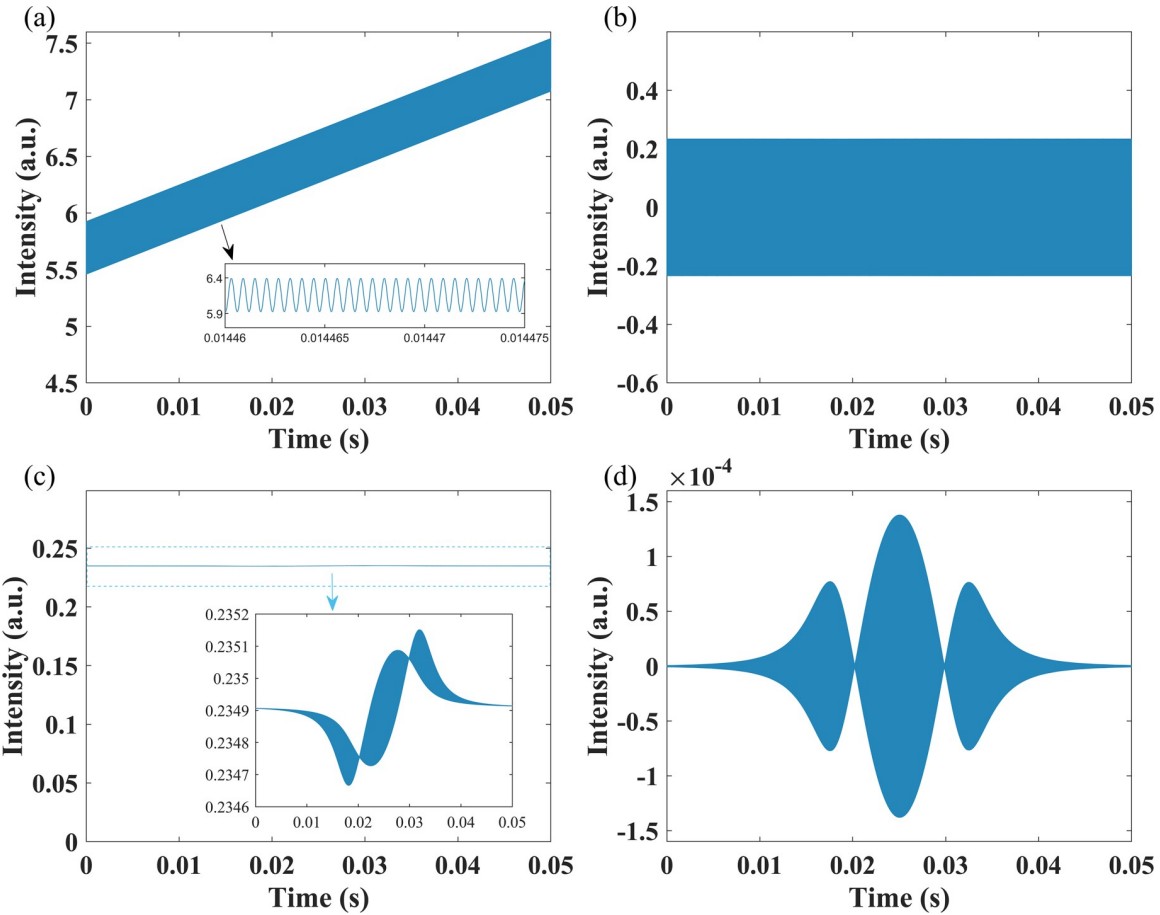

**Fig 2.** Signals in the signal processing of the proposed method: (a) $I_t(t)$, the transmitted light intensity signal; (b) $I(t)$, the band-pass filtered signal of $I(t)$; (c) $|z_1(t)|$, the envelope of $I(t)$; (d) $|z_1(t)|_{1f}$, the $1f$ component of $|z_1(t)|$.

As shown in Fig 2(D), $|z_1(t)|_{1f}$ presents a profile highly similar to the second harmonic. Finally, the demodulated signal, that is, the envelop of $|z_1(t)|_{1f}$ is extracted by performing Hilbert transform to $|z_1(t)|_{1f}$, and the result is presented in Fig 3.

Subsequently, the second harmonic demodulated by the proposed method is compared to that of lock-in amplification in Fig 4. Fig 4(A) presents the second harmonic demodulated by Hilbert transform based method and lock-in amplifier in red and blue, respectively, and the two signals are highly coincident and almost overlap. Fig 4(B) shows the relative error curve with a relative error of almost 0% at the center of the second harmonic. The lock-in amplifier demodulated signal is taken as the true value.

What stands out in Fig 4(B) is that the relative error increases gradually as the laser scans away from the center of the spectral line. This is due to the brick-wall bandpass filtering used in this simulation, which introduces a ringing artifacts error between the two signals. Fig 5 shows the error between the second harmonic demodulated by the two methods.

Therefore, the relative error increases gradually on both sides of the spectrum where the amplitude of the second harmonic gradually approaches zero. In spite of this, the data there is less important in practical applications. And satisfactory results have been obtained near the center of the spectral line.

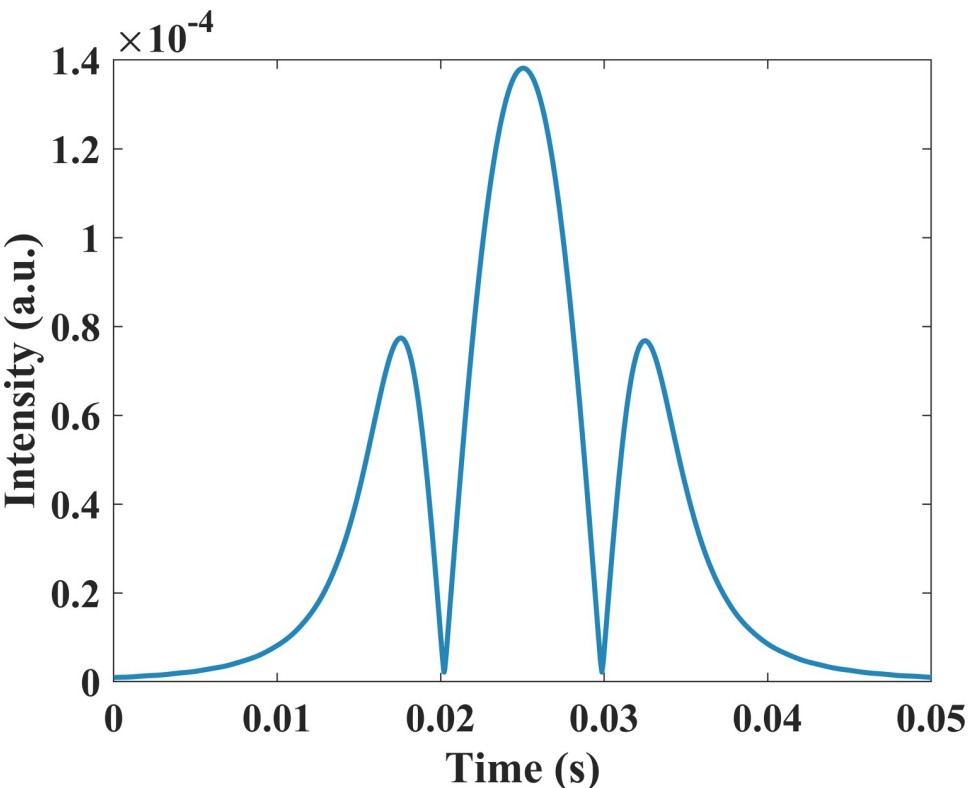

**Fig 3. The second harmonic demodulated by the Hilbert transform-based method.**

## 3.3 Error under different absorbance

Furthermore, the Hilbert transform based method under different absorbance conditions has been tested. Fig 6 shows the relationship curve for the relative error at the center of the second harmonic and $\alpha(\nu_0)$ (the absorbance at the line center). In Fig 6, the absorbance at the line center varies from 0% to 5% with the concentration of $CO_2$ ascend to 32.29%. And other parameters are same with Fig 4.

As shown in Fig 6, the relative error increases as the absorbance at the line center rises. The main reason is that with the increase of absorbance, the intensity of high order harmonics is

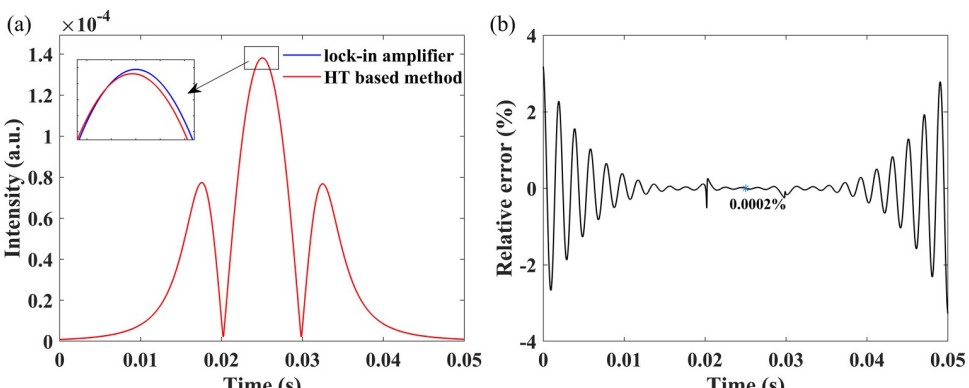

**Fig 4.** Simulation results at 400 ppm of $CO_2$: (a) Second harmonic demodulated by the two methods; (b) The relative error.

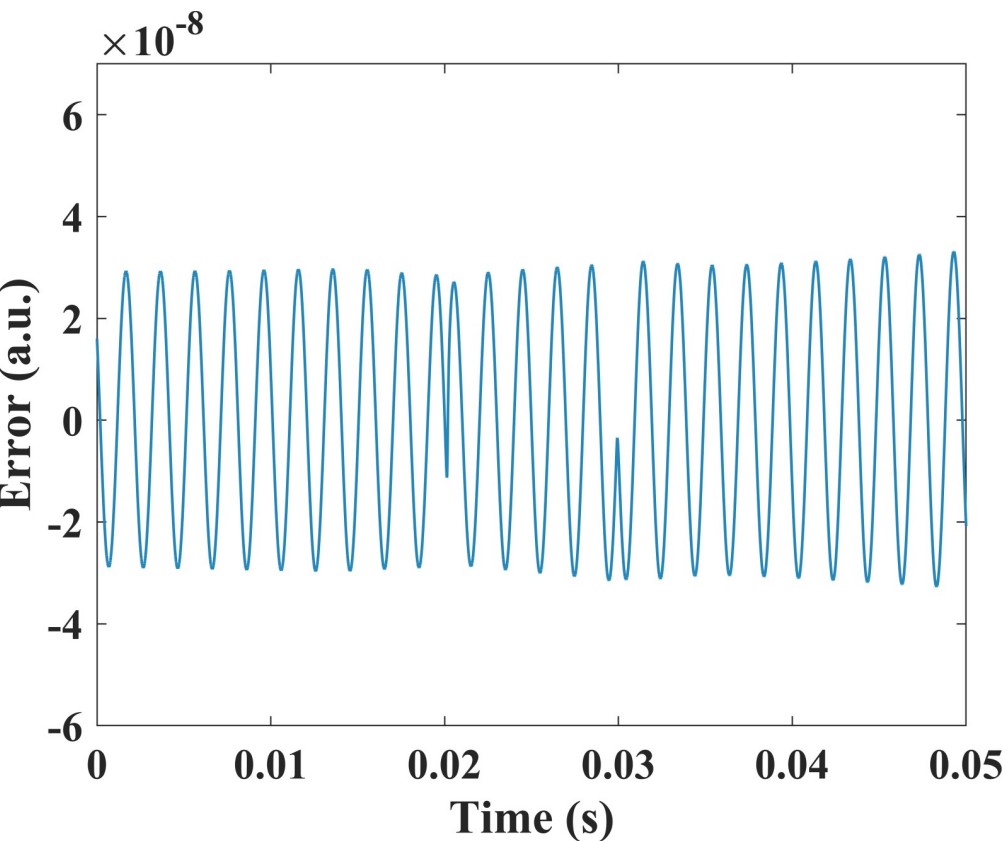

**Fig 5. The error between the second harmonic demodulated by the two methods.**

increasing relative to the first harmonic, and the approximation of Eq (16) is no longer satisfied. Nevertheless, the absolute relative error stay within 1% when the absorbance is no more than 3%, which indicated the proposed method is applicable to most trace gas detection cases.

## 4. Experiment

To verify the accuracy and validity of the proposed method, the second harmonic in a TDLAS signal obtained from an experiment is demodulated by the proposed method, and the result is compared with that of the lock-in amplification.

A DFB diode laser (nanoplus DFB TOP WL 2004.0 nm) is employed as the spectroscopic light source in the experiment. Its operating temperature is maintained at 40°C by a temperature controller. A laser driver provides the injection current of the DFB diode laser: a sawtooth current (10 Hz) with an amplitude of 5 mA and a sinusoidal current (21 kHz) with an amplitude of 3.5 mA are added to a DC injection current of 70 mA to produce the scanning signal and the modulation signal. The output frequency of the laser scans from 4992.19 $cm^{-1}$ to 4992.84 $cm^{-1}$, near the $CO_2$ absorption line at 4992.516 $cm^{-1}$ (wavelength of 2002.998 nm).

The length of the light path between the laser and the detector is 11mm. In the experiment, the gas in a bottle containing a little coke was detected, and the concentration of the carbon dioxide is uncertain, probably around 15%.

The TDLAS signal is detected by an InGaAs photodiode (LSIPD26-1) and collected by a digital oscilloscope (Rigol DS1102Z-E) with a sampling frequency of 25 MHz. Then, the transmitted light intensity signal shown in Fig 7 is processed by the proposed method and lock-in amplification on a computer.

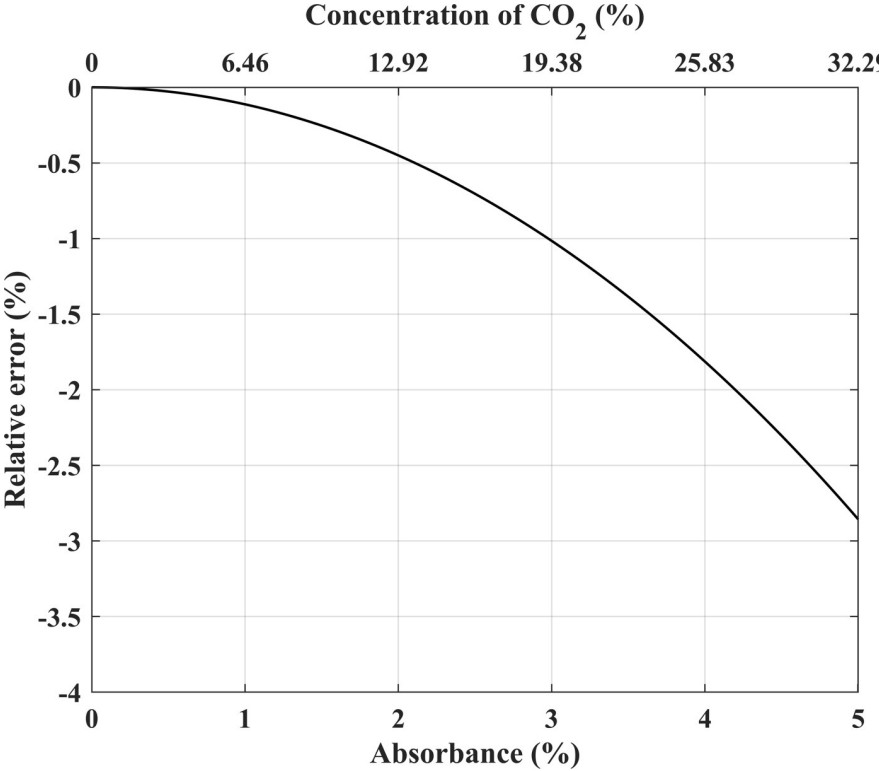

**Fig 6. The relative error at the center of the second harmonic under different absorbances.**

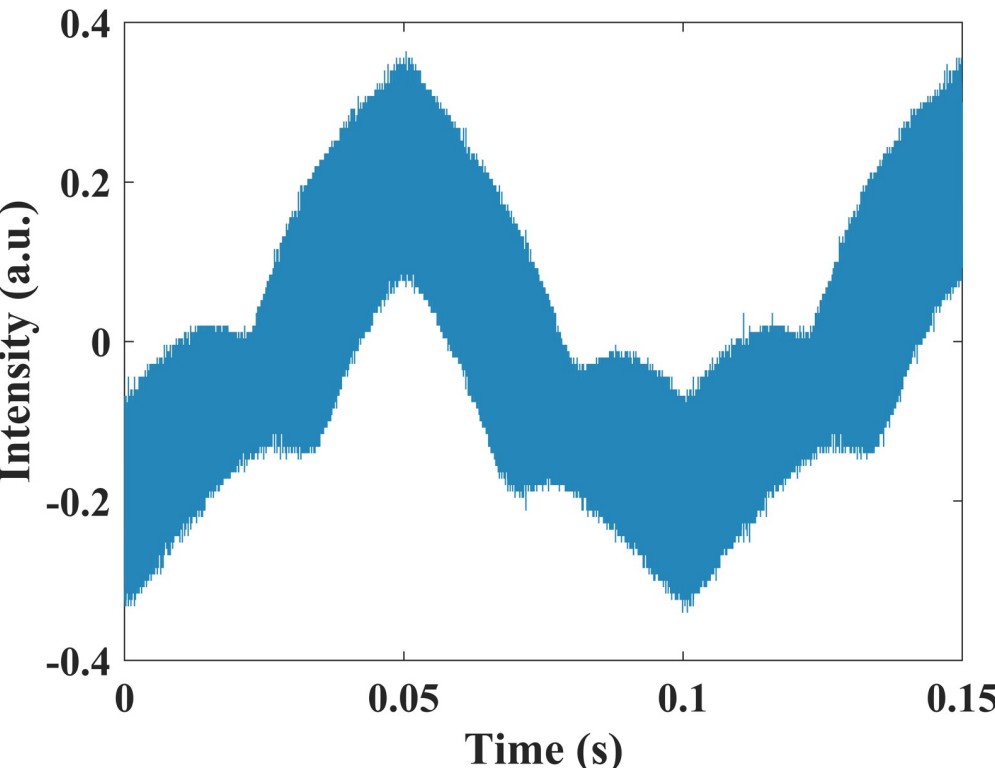

**Fig 7. The transmitted light intensity signal.**

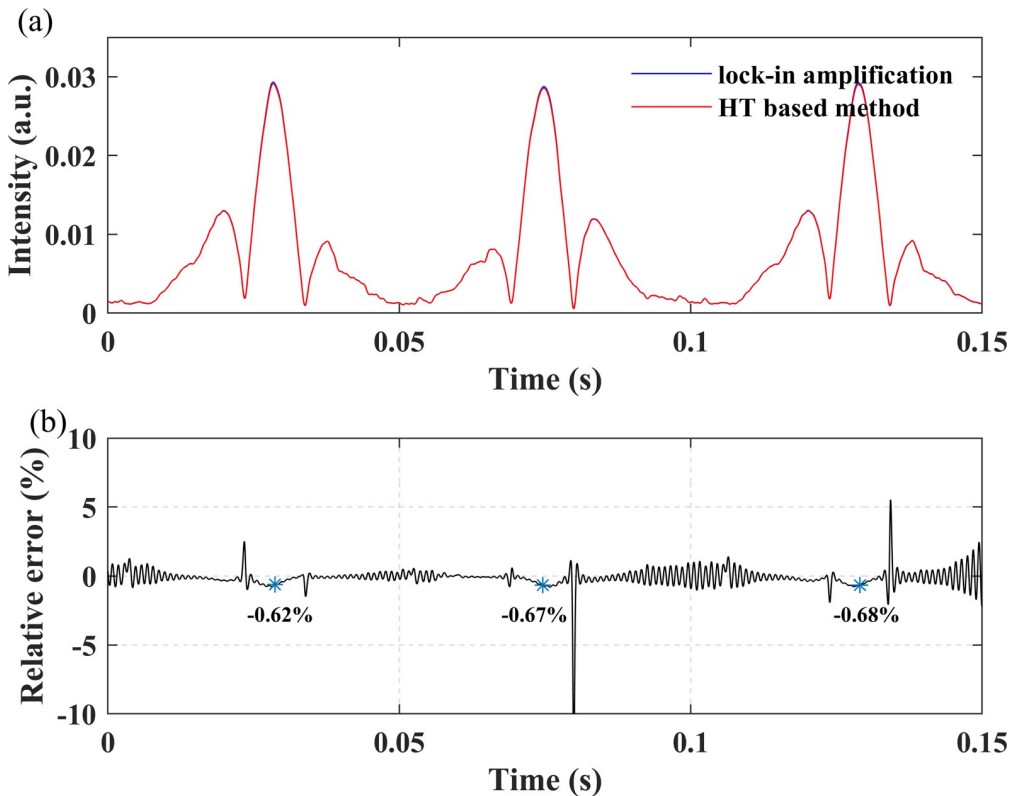

**Fig 8.** Experiment results: (a) Second harmonic demodulated by the two methods; (b) The relative error.

The modulation frequency $f$ in this experiment was 21 kHz, and in the program the cutoff frequency of the first band-pass filtering was set as [0.5*$f$, 2.5*$f$], which is [10500 Hz, 52500 Hz], and the cutoff frequency of the second band-pass filtering was set as [0.8*$f$, 1.2*$f$], which is [16800 Hz, 25200 Hz]. The stability of the proposed algorithm mainly depends on the SNR of the second harmonic signal. In the case of low SNR, the passband bandwidth of bandpass filtering should be set narrower.

The results of both methods are compared in Fig 8(A) and 8(B) shows the relative error of the proposed method versus lock-in amplification. The relative errors at the center of the spectral line are all marked out in blue in Fig 8(B). The results are similar to those of simulation and the signals demodulated by the two methods are almost overlap. Except near the local minimum of the second harmonic, the relative error of other positions is less than 2%, and the relative error at the center of the second harmonic is 0.6 ~ 0.7%.

## 5. Conclusion

A Hilbert based demodulation method for second harmonic in TDLAS technology is proposed. As no reference signal is required, the proposed method benefits second harmonic demodulation with convenient operation and efficiency. Theoretical analysis of this method shows that the error of this method is positively correlated with the intensity of the higher harmonics relative to the first harmonics, and this is verified by simulations. The errors under different absorbance are analyzed by simulation, and the absolute relative error stay within 1% when the absorbance is below 3%, which shows that the peak value of the second harmonic is approximately linear with the gas concentration at weak absorbance conditions, and the

proposed method is applicable to most trace gas detection cases. The accuracy of this method under low absorbance was verified by experiments and the codes and experiment data in this paper are all freely available as open source on GitHub (https://github.com/awublack/Hbt-TDLAS-secondHarmonic).

## Author Contributions

**Data curation:** Junfeng Wu.

**Funding acquisition:** Guohua Kang.

**Investigation:** Guohua Kang, Xu Li.

**Methodology:** Junfeng Wu.

**Software:** Hanyu Chen.

**Validation:** Xu Li.

**Visualization:** Xu Li.

**Writing – original draft:** Junfeng Wu, Hanyu Chen.

**Writing – review & editing:** Junfeng Wu, Hanyu Chen, Guohua Kang.

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
