## [Decision Letter · Decision Letter 0]

27 Sep 2022

PONE-D-22-22910TDLAS second harmonic demodulation based on Hilbert transformPLOS ONE

Dear Dr. Wu,

Thank you for submitting your manuscript to PLOS ONE. After careful consideration, we feel that it has merit but does not fully meet PLOS ONE’s publication criteria as it currently stands. Therefore, we invite you to submit a revised version of the manuscript that addresses the points raised during the review process.

We look forward to receiving your revised manuscript.

Kind regards,

Xuejian Wu, Ph.D.

Academic Editor

PLOS ONE

Journal Requirements:

Reviewers' comments:

Reviewer's Responses to Questions

**Comments to the Author**

1. Is the manuscript technically sound, and do the data support the conclusions?

Reviewer #1: Partly

Reviewer #2: Yes

Reviewer #3: Yes

2. Has the statistical analysis been performed appropriately and rigorously? 

Reviewer #1: Yes

Reviewer #2: Yes

Reviewer #3: N/A

3. Have the authors made all data underlying the findings in their manuscript fully available?

Reviewer #1: Yes

Reviewer #2: No

Reviewer #3: No

4. Is the manuscript presented in an intelligible fashion and written in standard English?

Reviewer #1: Yes

Reviewer #2: Yes

Reviewer #3: No

5. Review Comments to the Author

Reviewer #1: 1. In the Experiment section, the optical path is filled with gas in a Coke bottle, and second harmonic is gotten with two methods that of LIA and Hilbert transform. By comparing the error of the two harmonics obtained by the two methods, the Hilbert tansform method is validated.However, just as the paper said, the experiment is only conducted under one concentration condition of 15%. Therefore, the conclusion is not convincing enough.Please Supply the experiments and results under other concentration conditions.

2. When comparing the difference of the two methods, only the error of harmonic is analysed.To describe the full performance of the method, some other performance evaluations should be vaiidated, such as algorithm speed、stability and so on.

3.Please supply some experimental details,like the cutoff frequecy of band-pass filtering.

4.Please supply a physical picture of the experiment.

Reviewer #2: The authors demodulated using the Hilbert transform to demodulate the second harmonic signal of TDLAS without a reference signal. they validate the method by simulation and experiment, and the result shows that the method is applicable to practical trace gas detection. It sounds interesting and plausible.

1) the author declared "The concentration of CO2 in the atmosphere is close to 400 ppm" in Line 189. It is not true for ambient CO2 concentration being much higher.

2) A clear explanation for every panels of Fig. 2 is necessary.

3) An explanation or discussion for the "error increases as the laser scans away from the center of the spectral line" is necessary.

4) A detailed experiment setup is necessary, such as the product model of digital acquisition, lock-in amplifier used in the manuscript, and their parameters setting.

Reviewer #3: A demodulation method for tunable diode laser absorption spectroscopy (TDLAS) second harmonic based on the Hilbert transform is proposed in this paper. But, Hilbert transform is a classical method and has been used to obtain an analytic representation in the field of gas detection. So, it is recommended that authors should not shy away from the current Hilbert transform methods, and proposed an improved Hilbert transform to process the gas data with better performance. Similar references are as following:

1. Efficient terahertz absorption gas sensor with Gaussian process regression in time- and frequency-domain, Sensors and Actuators B: Chemical, 2022.

2. Computational Doppler-limited dual-comb spectroscopy with a free-running all-fiber laser, APL Photonics, 2019.

6. PLOS authors have the option to publish the peer review history of their article (what does this mean?). If published, this will include your full peer review and any attached files.

Reviewer #1: No

Reviewer #2: No

Reviewer #3: No

---

## [Author Response · Author response to Decision Letter 0]

18 Oct 2022

Response to Reviewer #1

1) In the Experiment section, the optical path is filled with gas in a Coke bottle, and second harmonic is gotten with two methods that of LIA and Hilbert transform. By comparing the error of the two harmonics obtained by the two methods, the Hilbert transform method is validated. However, just as the paper said, the experiment is only conducted under one concentration condition of 15%. Therefore, the conclusion is not convincing enough. Please Supply the experiments and results under other concentration conditions.

Response: We agree with your comment that further experiments under other concentration conditions will help to reinforce our conclusions. Your suggestion provides a direction for our next research. Unfortunately, the research project has ended, and the experimental equipment we made has been delivered. At present, we do not have sufficient conditions to support further supplementary experiments.

2) When comparing the difference of the two methods, only the error of harmonic is analysed. To describe the full performance of the method, some other performance evaluations should be validated, such as algorithm speed, stability and so on.

Response: In terms of the speed of the algorithm, for the same data, the processing time of the proposed algorithm is about 5 times that of the lock-in amplification method, and the main processing time is spent on filtering and calculating the Hilbert transform of the signal; Compared with lock-in amplification method, the proposed method is more sensitive to noise, so band-pass filtering is used in this paper to improve the stability of the algorithm against noise.

3) Please supply some experimental details, like the cutoff frequency of band-pass filtering.

Response: Our data is processed on a computer. The modulation frequency f in our experiment was 21000 Hertz, so in the program the cutoff frequency of the first band-pass filtering was set as [0.5*f, 2.5*f], which is [10500 Hz, 52500 Hz], and the cutoff frequency of the second band-pass filtering was set as [0.8*f, 1.2*f], which is [16800 Hz, 25200 Hz]. A detailed description of experimental including the cutoff frequency of band-pass filtering has been added to the revised manuscript. See line 278-281 of the revised manuscript.

4) Please supply a physical picture of the experiment.

Response: 

Adding images is not supported here, and the physical picture of the experiment can be seen in the attach file labeled 'Response to Reviewers'.

In the experiment, the gas in a bottle containing a little coke was detected. It has been corrected in the revised manuscript.

Original text (Line 250): In the experiment, this section of the optical path is filled with gas in a Coke bottle.

Modified text (Line 272): In the experiment, the gas in a bottle containing a little coke was detected.

Response to Reviewer #2

1) The author declared "The concentration of CO2 in the atmosphere is close to 400 ppm" in Line 189. It is not true for ambient CO2 concentration being much higher.

Response: The statement here is not rigorous enough. It has been corrected in the revised manuscript.

Original text (Line 189): The concentration of CO2 in the atmosphere is close to 400 ppm.

Modified text (Line 190): Global average atmospheric carbon dioxide is about 400 ppm.

2) A clear explanation for every panel of Fig. 2 is necessary.

Response: A clear explanation for every panel of Fig. 2 has been included in the revised manuscript.

(Line 202-214): The transmitted light signal in Fig. 2(a) consists of low-frequency sawtooth wave and high-frequency harmonics (the figure shows the signal for only half of the scanning period), which are the common results of the intensity modulation of injection current and nonlinear absorption; In Fig. 2(b), all signals except the first harmonic (1f) and second harmonic (2f) are filtered out, corresponding to Eq. (8); Fig. 2(c) is the envelope of the signal in Fig. 2(b). Since the intensity modulation leads to a strong gas-independent RAM signal in the first harmonic, the envelope change in the figure is not obvious, the gas related signal can be seen by zooming on the vertical axis and looks like a lying “S” plus some high-frequency component. These high-frequency component is the second harmonic signal that is transformed to 1f, corresponding to Eq. (16); Then the 1f component of |z1(t)| is obtained by band-pass filtering and presented in Fig. 2(d).

3) An explanation or discussion for the "error increases as the laser scans away from the center of the spectral line" is necessary.

Response: An explanation for the "error increases as the laser scans away from the center of the spectral line" has been included in the revised manuscript.

(Line 230-239): What stands out in Fig. 4 (b) is that the relative error increases gradually as the laser scans away from the center of the spectral line. This is due to the brick-wall bandpass filtering used in this simulation, which introduces a ringing artifacts error between the two signals. Fig. 5 shows the error between the second harmonic demodulated by the two methods.

Fig. 5. The error between the second harmonic demodulated by the two methods.

Therefore, the relative error increases gradually on both sides of the spectrum where the amplitude of the second harmonic gradually approaches zero.

4) A detailed experiment setup is necessary, such as the product model of digital acquisition, lock-in amplifier used in the manuscript, and their parameters setting.

Response: Our raw data were collected using an oscilloscope with a sampling frequency of 25MHz, and all the processing of the raw data described in this paper was carried out on a computer, so the lock-in amplifier was implemented by programming. The following figure shows our experimental setup.

The circuit board is a lock-in amplifier designed by ourselves, but the data from this lock-in amplifier is not used in this paper.

A detailed experiment setup such as the product model of digital acquisition has been included in the manuscript.

(Line 274-276): The TDLAS signal is detected by an InGaAs photodiode (LSIPD26-1) and collected by a digital oscilloscope (Rigol DS1102Z-E) with a sampling frequency of 25 MHz.

Response to Reviewer #3

1) A demodulation method for tunable diode laser absorption spectroscopy (TDLAS) second harmonic based on the Hilbert transform is proposed in this paper. But, Hilbert transform is a classical method and has been used to obtain an analytic representation in the field of gas detection. So, it is recommended that authors should not shy away from the current Hilbert transform methods, and proposed an improved Hilbert transform to process the gas data with better performance. Similar references are as following:

1. Efficient terahertz absorption gas sensor with Gaussian process regression in time- and frequency-domain, Sensors and Actuators B: Chemical, 2022.

2. Computational Doppler-limited dual-comb spectroscopy with a free-running all-fiber laser, APL Photonics, 2019.

Response: Hilbert transform is a classical method and has been used to obtain an analytic representation in the field of gas detection. But these two articles you recommend have little relevance to the topic of this paper. To the best of the authors' knowledge, the second harmonic demodulation method proposed in this paper has not been proposed in the previous literature. However, our scope of knowledge is limited and if you know of any other papers that have proposed a similar approach, you are welcome to recommend them to us.

---

## [Decision Letter · Decision Letter 1]

16 Nov 2022

PONE-D-22-22910R1TDLAS second harmonic demodulation based on Hilbert transformPLOS ONE

Dear Dr. Wu,

Thank you for submitting your manuscript to PLOS ONE. After careful consideration, we feel that it has merit but does not fully meet PLOS ONE’s publication criteria as it currently stands. Therefore, we invite you to submit a revised version of the manuscript that addresses the points raised during the review process.

We look forward to receiving your revised manuscript.

Kind regards,

Xuejian Wu, Ph.D.

Academic Editor

PLOS ONE

Journal Requirements:

Reviewers' comments:

Reviewer's Responses to Questions

**Comments to the Author**

1. If the authors have adequately addressed your comments raised in a previous round of review and you feel that this manuscript is now acceptable for publication, you may indicate that here to bypass the “Comments to the Author” section, enter your conflict of interest statement in the “Confidential to Editor” section, and submit your "Accept" recommendation.

Reviewer #1: (No Response)

Reviewer #2: All comments have been addressed

2. Is the manuscript technically sound, and do the data support the conclusions?

Reviewer #1: Partly

Reviewer #2: Yes

3. Has the statistical analysis been performed appropriately and rigorously? 

Reviewer #1: Yes

Reviewer #2: Yes

4. Have the authors made all data underlying the findings in their manuscript fully available?

Reviewer #1: Yes

Reviewer #2: Yes

5. Is the manuscript presented in an intelligible fashion and written in standard English?

Reviewer #1: Yes

Reviewer #2: Yes

6. Review Comments to the Author

Reviewer #1: The article has been well revised, but I still have some questions.

1. Why only one band-pass filter is used in the introduction of Hilbert transform method in FIG. 1, while two band-pass filters are used in the experiment, which are inconsistent with each other?

2. In this paper, the second harmonic signal obtained by phase-locked amplification method is taken as the real signal, and compared with the second harmonic signal obtained by Hilbert transform method, so as to show the relative error of Hilbert transform method. However, the second harmonic obtained by the phase-locked amplification method is also different according to its different filter parameters. At this time, the error of the signal will change when the two methods are compared. Then, is the phase-locked amplification method used in this paper the optimal effect of demodulation of the second harmonic (that is, are the relevant parameters of the phase-locked amplification process best)? If not, then the error analysis will lose some credibility. Please elaborate on this question.

3. Is it more convincing to take the stability of the second harmonic signal obtained by the Hilbert transform method and the ability to characterize the concentration of the gas to be measured as the evaluation basis of the performance index of this method instead of just comparing with the phase-locked amplification method?

4. In line 286 to 287, in order to express the similar results obtained by the two methods, describing specific performance indicators may be better than "almost overlap".

5. The scanning mode in this paper is triangular wave scanning. It can be seen from Figure 8b that the second harmonic signal corresponding to the rising part of triangular wave and the signal corresponding to the falling part seem to show different error characteristics. Is it better to switch to a sawtooth scan?

Reviewer #2: The manuscript has been revised according to the comments of the reviewers. I recommend to accept it for publication.

7. PLOS authors have the option to publish the peer review history of their article (what does this mean?). If published, this will include your full peer review and any attached files.

Reviewer #1: No

Reviewer #2: No

---

## [Author Response · Author response to Decision Letter 1]

21 Nov 2022

Response to Reviewer #1

1) Why only one band-pass filter is used in the introduction of Hilbert transform method in FIG. 1, while two band-pass filters are used in the experiment, which are inconsistent with each other?

Response: The third step “1f component extraction” in FIG 1 is actually implemented by bandpass filtering as well. We mentioned this in line 155 and 156 of the revised manuscript of the last edition. And considering that “1f component extraction” can better express the practical significance of this step than simply calling it “bandpass filtering”, this paper refers to this step as “1f component extraction”. The revised manuscript has made a supplementary explanation for this.

2) In this paper, the second harmonic signal obtained by phase-locked amplification method is taken as the real signal, and compared with the second harmonic signal obtained by Hilbert transform method, so as to show the relative error of Hilbert transform method. However, the second harmonic obtained by the phase-locked amplification method is also different according to its different filter parameters. At this time, the error of the signal will change when the two methods are compared. Then, is the phase-locked amplification method used in this paper the optimal effect of demodulation of the second harmonic (that is, are the relevant parameters of the phase-locked amplification process best)? If not, then the error analysis will lose some credibility. Please elaborate on this question.

Response: The lock-in amplifier was implemented by programming. We used orthogonal vector lock-in amplifier to avoid distortion caused by inaccurate phase of reference signal, and therefore the main factors affecting its performance are reference signal frequency and filter parameters.

The frequency of the reference signal was set as 42000 Hz, which is twice the frequency of modulation frequency (that is, equal to the frequency of the second harmonic). And in order to speed up the filtering process, the inverse Fourier transform is used in the programming script to implement a brick-wall low-pass filtering. When setting the filtering cutoff frequency, we ensured that the second harmonic signal can be restored as much as possible. The following two figure shows the amplitude spectrum of the pre-filtered and post-filtered signals in two orthogonal vector “sin” and “cos”,

Fig. 1. The amplitude spectrum of the pre-filtered and post-filtered signal in “sin” vector.

Fig. 2. The amplitude spectrum of the pre-filtered and post-filtered signal in “cos” vector.

(Figures is not supported here, you can view them in attachment "Response to Reviewers.docx")

We found that the energy of the second harmonic line mainly concentrated in the frequency below 200 Hz, and there was also a small amount of energy between 200-500Hz. Therefore, we set the cut-off frequency as 1000 Hz to avoid the distortion of the second harmonic waveform caused by filtering as much as possible, and also filtered out the high-frequency noise.

3) Is it more convincing to take the stability of the second harmonic signal obtained by the Hilbert transform method and the ability to characterize the concentration of the gas to be measured as the evaluation basis of the performance index of this method instead of just comparing with the phase-locked amplification method?

Response: The main purpose of this paper is the demodulation of the second harmonic, and the ability to characterize gas concentrations is a secondary application of this method. In the case of low absorbance, the Hilbert transform method can be used to demodulate the second harmonic signal. Therefore, the second harmonic obtained by the proposed method is compared with that of the lock-in amplifier method. 

The stability of the proposed algorithm mainly depends on the SNR of the second harmonic signal. In the case of low SNR, the passband bandwidth of bandpass filtering should be set narrower.

4) In line 286 to 287, in order to express the similar results obtained by the two methods, describing specific performance indicators may be better than "almost overlap".

Response: The statement here has been revised in the manuscript.

Original text (Line 286-287): Similar to the simulation results, the signals demodulated by the two methods are almost overlap.

Modified text (Line 290-293): The results are similar to those of simulation and the signals demodulated by the two methods are almost overlap. Except near the local minimum of the second harmonic, the relative error of other positions is less than 2.5%, and the relative error at the center of the second harmonic is 0.6 ~ 0.7%.

5) The scanning mode in this paper is triangular wave scanning. It can be seen from Figure 8b that the second harmonic signal corresponding to the rising part of triangular wave and the signal corresponding to the falling part seem to show different error characteristics. Is it better to switch to a sawtooth scan?

Response: The scanning mode is not the main factor affecting the error, but switching to a sawtooth scan does provide better signal consistency (the waveform is basically the same from every scan).

---

## [Editor Report · Decision Letter 2]

22 Nov 2022

TDLAS second harmonic demodulation based on Hilbert transform

PONE-D-22-22910R2

Dear Dr. Wu,

We’re pleased to inform you that your manuscript has been judged scientifically suitable for publication and will be formally accepted for publication once it meets all outstanding technical requirements.

Kind regards,

Xuejian Wu, Ph.D.

Academic Editor

PLOS ONE
---

## [Editor Report · Acceptance letter]

28 Nov 2022

PONE-D-22-22910R2 

TDLAS second harmonic demodulation based on Hilbert transform 

Dear Dr. Wu:

I'm pleased to inform you that your manuscript has been deemed suitable for publication in PLOS ONE. Congratulations! Your manuscript is now with our production department. 

Kind regards, 

on behalf of

Dr. Xuejian Wu 

Academic Editor

PLOS ONE